# "Healing hands: Understanding physiotherapists' knowledge, attitudes and barriers on pressure injury prevention in the medical wards in Jordan"

Enas A. Assaf[1]*, Rahaf Alkhresheh[2], Taher Assaf[3], Suhair Al-Ghabeesh[4], Ramasubbamma Ramaiah[1]

1 College of Nursing, Khamis Mushait, King Khalid University Abha, Saudia Arabia, 2 Faculty of Nursing, Applied Science Private University, Amman, Jordan, 3 Faculty of Agriculture, Jerash University, Jerash, Jordan, 4 Faculty of Nursing, Al-Zaytoonah University of Jordan, Amman, Jordan

* eassaf@kku.edu.sa

## Abstract

### Aim

to highlight the physiotherapist's knowledge and attitudes, as well as barriers and related factors affecting Pressure Injury prevention in the medical wards of Jordanian hospitals.

### Method

Cross-sectional correlational study of all physiotherapists working in the three largest governmental hospitals in Jordan (north, capital, south), focusing on medical ward services. A validated instrument for measuring knowledge and attitudes towards pressure injury prevention, as well as selected barriers identified in the literature, was distributed for self-administration. Descriptive analysis, a t-test for two-item variables, and ANOVA (F-value) for variables containing more than two items. Multivariate regression analysis was employed to assess the relationship between the independent variables (demographic factors) and the outcome variables (knowledge, attitude, and perceived barriers). Differences were considered statistically significant at a level of $P < 0.05$.

### Results

Overall, physiotherapists demonstrate a good understanding and positive attitudes towards the prevention of pressure injuries. However, barriers were marked in the study as the majority considered shortage of resources (94.4%), lack of multidisciplinary team among the health care team (88.9) and uncooperative patients (88.9%) followed by presence of other priorities at work pressure injury prevention (83.3%), and in adequate knowledge (80.6%). Monthly income, working experience, training, and using guidelines were statistically significant predictors of the knowledge score (p values: 0.01, 0.03, 0.02, 0.001, respectively). Age was statistically significant in

**Data availability statement:** The data underlying the results presented in the study are available from the supporting files, as we uploaded the excel file for the data.

**Funding:** This research was funded by the Deanship of Scientific Research at King Khalid University; Grant number "RGP 2/263/46".

**Competing interests:** The authors have declared that no competing interests exist.

relation to the attitude score (P < 0.001). None of the demographic characteristics were significantly related to the perceived barriers.

## Conclusion

Physiotherapists can play a significant role in preventing pressure injuries when they possess improved knowledge, as evidenced by their positive attitudes towards preventing pressure injuries for patients in medical wards. However, more attention is recommended to allocate resources and staff, as well as implement policy changes, to enhance a multidisciplinary team approach in working with medical wards, addressing their special needs, overcrowding, and comorbidity among patients. Further research is recommended to assess the roles and barriers faced by physiotherapists in medical wards and other areas.

## Introduction

Pressure Injury [1] is described as localized damage caused by pressure, or pressure combined with shear, to the skin and/or underlying tissue, typically over a bony prominence [2]. Despite technological advancements and preventive measures, PI remains a serious global health concern. The prevalence of PI was 12.8% globally, 14.5% in Europe, 13.6% in North America, 12.7% in South America, 3% in Asia, 12.6% in the Middle East, and 9% in Australia between 2008 and 2018 [3,4]. A lack of knowledge, insufficient skills, and a negative attitude among healthcare providers about preventing PI essentially cause the deterioration of PI in medical wards [5,6]. Surrounding Arab countries were studied to determine the incidence rate of PIs in Kuwait. Seven health regions oversee the various tiers of care provided by Kuwait's public health system. As a result, 203 patients (17.1%) with PIs were discovered out of 1,186 patients hospitalized in 54 medical wards at Kuwait's seven leading general hospitals [4]. In Egypt, PI was found to affect 10–25.9% of seriously ill patients confined to intensive care units [7]. According to a Saudi Arabian study, the prevalence and incidence of acute care PI were 44.4% and 38.6%, respectively [8].

A recent study in Jordan by Najjar et al. [9] discussed PIs related to devices such as nasogastric tubes, face masks, Pulse oximetry probes, and intravenous catheters, which were associated with almost half of the injuries. The study shows that severe risk was present in about half (46.3%) of the patients. Merely 17% of patients who posed a danger received appropriate preventative measures.

Pressure injuries are a significant concern in medical wards in Jordan because they can lead to pain, infection, and delayed healing. Proper assessment and management are crucial for preventing these injuries and providing the best care for patients. Healthcare professionals often implement strategies such as regular skin assessments, turning schedules, proper nutrition, and pressure-relieving devices to address the issue effectively [10,11]. A multidisciplinary approach involving nurses, physicians, physiotherapists, and wound care specialists is essential to tackle PI comprehensively [12,13].

Physiotherapists can assist in developing personalized mobility and positioning plans to reduce pressure on vulnerable areas [14]. The importance of physiotherapy in treating PI is magnified. These therapies may include teaching caregivers, strengthening exercises, preventing wounds, and transferring from a bed to a chair. PI can be prevented by shifting positions cyclically, meaning cleaning guidelines and time frames should be followed consistently after each exercise routine [15]. A physiotherapist's job in managing PI is to help a patient get mobile and monitor their skin conditions and function through appropriate placement [16–18]. A physiotherapist evaluates a patient's independence, cognitive function, bed mobility, transitional movements, transfers, and activities of daily living function, it is possible to observe the need to monitor previous physical health state and take certain precautions related to procedures [16]. A study in Multan evaluated the knowledge and attitudes of 130 physical therapists regarding pressure injuries (PIs) across government and private hospitals. Using the Pressure Injury Knowledge Test (PIKT) and the Scale of Attitudes Toward PI Prevention, the results of the study showed that most therapists had satisfactory knowledge and positive attitudes towards PI prevention [14].

Prioritizing PI prevention in Jordanian hospital wards may be complicated by conflicting priorities, limited resources, and a decreased sense of PI danger in comparison to critical care units [19]. Large patient ratios, scarce resources, and high workloads are common in medical wards, which results in a reactive rather than proactive approach to PI care, with particular interventions being given only after accidents occur [20]. Therefore, a multidisciplinary team approach is highly recommended for preventing PI. Studying the role, knowledge, and attitudes, as well as barriers toward PI, among physiotherapists is limited in medical wards. Therefore, this study aimed to assess physiotherapists' knowledge, attitudes, and perceived barriers toward pressure injury prevention in Jordanian medical wards and identify associated demographic factors.

## Maternal and method

### Study design and sampling

A descriptive correlational cross-sectional study was conducted involving all 36 physiotherapists employed in the physiotherapy departments and directly engaged in patient care within the medical wards of the three largest governmental hospitals in the Capital, Amman, as well as in Irbid in the north and Al-Karak in the south of Jordan. The sample was obtained through convenience sampling, which is defined as selecting the most readily available healthcare providers as participants in a study [21]. Data was collected in two months between (20/08–20/10/2024)

These hospitals were chosen because they serve patients from diverse socio-economic backgrounds, accommodating individuals from all governorates and neighboring regions, and they contain the largest physiotherapy departments among governmental facilities. The inclusion criteria encompassed all physiotherapists working in these hospitals, irrespective of their educational qualifications. Given that the responsibilities of physiotherapists in government hospitals are consistent regardless of their educational level, those in administrative roles who do not work directly with patients were excluded from the study.

The sample size is defined as the number of participants needed to achieve statistically reliable results [22]. To determine the appropriate sample size for this research, the researchers employed G*power software for analysis. They set an alpha level of 0.05, a power of 95%, and a mild effect size of 0.15 for a two-tailed test. The Pearson Correlation was chosen as the statistical method, as it offers a standardized measure of the linear relationship between the variables.

### Research tool

The study used a self-administered questionnaire [23]. The language of the questionnaire was originally written in English and it was administered in the same language as there were no limitations for healthcare providers in reading or writing English language. The questionnaire was divided into four sections. The first section collected demographic information and related factors from healthcare providers in the medical wards of Jordanian hospitals. It included data on gender, age, marital status, qualifications, monthly income, work experience, training on pressure injuries (PIs), adherence to pressure

injury guidelines, and perceptions of inadequate staffing. The tool demonstrated good convergent validity and a high level of internal consistency, with a reliability coefficient of 0.88 [23].

The second section assessed knowledge using an adapted, validated, and reliable questionnaire consisting of 18 questions. Each question offered three response options on a scale from 0 to 2, where 0 indicated "I do not know," 1 represented "False," and 2 signified "True." This tool also exhibited good convergent validity and a high degree of internal consistency, with a reliability coefficient of 0.88 [23].

The third section evaluated the attitudes of healthcare providers using the Pressure Injury Attitude Questionnaire, which comprised 11 questions rated on a 5-point Likert scale. Responses ranged from "5 – strongly agree" to "1 – strongly disagree." An attitude score equal to or above the mean of the attitude-related questions was considered a good attitude, while scores below the mean were deemed poor. This tool showed internal consistency reliability (Cronbach's α) of 0.76 [24].

The fourth section of the questionnaire included 12 closed-ended questions that required "Yes" or "No" responses to identify the barriers faced by physiotherapists in implementing pressure injury prevention protocols. These questions were adapted from a review of various pieces of literature [2,25,26].

A pilot study with 5 physiotherapists was done to ensure clarity of questions and appropriateness of the questionnaire from a cultural perspectives, and there were no difficulties, and we decided in include them in the study.

## Statistical analysis

The Data from the questionnaire were first cleaned and checked for completeness and consistency, and then they were entered into SPSS version 26 (IBM, New York) for analysis. All 36 participants completed the questionnaire in full; therefore, no missing data were present in the dataset. Descriptive statistics were used to summarize demographic, knowledge, attitude, and perceived barrier variables by using frequencies and percentages. A knowledge score for each case was calculated as the total number of correct answers out of the 18 items that measure PI prevention knowledge; the knowledge scores range from 0 to 18. To calculate the attitude scores, numerical values were assigned to each attitude question: 1 for "strongly disagree," 2 for "disagree," 3 for "neither agree nor disagree," 4 for "agree," and 5 for "strongly agree." The attitude score was calculated as the total points received for the 11 attitude answers and ranged from 11 to 55. Similarly, the perceived barrier score was calculated as the total number of questions answered "Yes" by the respondents, ranging from 0 to 11, corresponding to the 11 barrier items presented in the questionnaire.

To measure the relationship between the scores of the knowledge, attitude, and perceived barriers, the Pearson correlation was used. ANOVA analysis was used to determine the effect of qualification on the scores of knowledge, attitude, and perceived barriers. Knowledge, attitude, and perceived barriers were compared by demographic information for each qualification (physician, registered nurse, physiotherapist, and wound nurse) using a t-test for two-item variables and ANOVA (F-value) for variables containing more than two items. Multivariate regression analysis was employed to assess the adjusted relationship between the independent variables (demographic factors) and the outcome variables (knowledge, attitude, and perceived barriers), thereby controlling for the potential confounding effects of these factors. Differences were considered statistically significant at a level of $P < 0.05$.

## Ethical considerations

Ethical approval was granted by the Ministry of Health through the Institutional Review Board (IRB no. mba/ethical committee/12724) and by the Applied Science Private University (IRB no. 2023-2024-32). All physiotherapists working in the hospitals were invited to participate, along with a comprehensive explanation of the research objectives, significance, and duration. After receiving this information, participants were asked to sign an informed consent form, confirming their voluntary participation and their right to withdraw from the study at any time. To maintain confidentiality and anonymity, participants were instructed not to disclose personal information, including their names, contact details, or identification documents. All collected data were securely stored in a locked cabinet, accessible only to the researchers.

## Results

### Demographic characteristics

Table 1. Presents the demographic characteristics of the participants, with 22 physiotherapists being male and 14 being female, aged between 30 and 40, representing a percentage of 56.8%. Most of them are married (78.4%), while half of them receive a monthly salary of less than 500 JOD. 88% have more than ten years of experience, and approximately 61.1% have received PI prevention training, but it is older than two years. The physiotherapists are divided into two groups: 22 (61.1%) used PI prevention guidelines, and 14 (38.9%) did not use PI prevention guidelines. Half of them stated that there is an unproportional provider-to-patient ratio.

### Knowledge, attitudes, and barriers of pressure injury

Table 2 shows that physiotherapists' knowledge of preventing PI is good, as evidenced by their selection of 'true' for all statements, with frequency values ranging from 94.4 to 52.8. A total of 35 physiotherapists highly agreed that risk factors for PI development are immobility, incontinence, impaired nutrition, and altered level of consciousness. However, only 19 of them agreed with statement number four, "Hot water and soap may dry the skin and increase the risk for PI."

**Table 1. Sociodemographic characteristics of the physiotherapist (n = 36).**

| Variable | Physiotherapist Frequency (%) |
|---|---|
| Sex | |
| ● Male | 22 (61.1) |
| ●Female | 14 (38.9) |
| Age | |
| ● < 30 | 7 (19.4) |
| ● 30-40 | 22 (61.1) |
| ● > 40 | 7 (19.4) |
| Marital<br>● Married<br>● Single | 28 (77.8)<br>8 (22.2) |
| Monthly Income<br>● ≤ 500<br>● > 500 | 18 (50.0)<br>18 (50.0) |
| Experience<br>● ≤ 5<br>● 5–10<br>● 11 - 15<br>● > 15 | 5 (13.9)<br>4 (11.1)<br>15 (41.7)<br>12 (33.3) |
| Received Training<br>● Yes<br>● No | 22 (61.1)<br>14 (38.9) |
| Training Date<br>● No<br>● Old | 14 (38.9)<br>22 (61.1) |
| Using Guidelines<br>● Yes<br>● No | 22 (61.1)<br>14 (38.9) |
| Unproportioned Providers<br>● Yes<br>● No | 18 (50.0)<br>18 (50.0) |

**Table 2. Physiotherapists' Knowledge towards Preventing PI (n = 36).**

| Variables | Physiotherapist's Knowledge | | |
|---|---|---|---|
| | True n (%) | False n (%) | Don't Know n (%) |
| 1. Risk factors for development of PI are immobility, incontinence, impaired nutrition, and altered level of consciousness | 34 (94.4) | 1 (2.8) | 1 (2.8) |
| 2.All hospitalized individuals at risk for PI should have a systematic skin inspection at least daily and those in long-term care at least once a week. | 28 (77.8) | 3 (8.3) | 5 (13.9) |
| 3.The first sign of PI development is redness or open sore. | 31 (86.1) | 3 (8.3) | 2 (5.6) |
| 4. Hot water and soap may dry the skin and increase the risk for PI. | 19 (52.8) | 11 (30.6) | 6 (16.7) |
| 5.It is important to massage bony prominences. | 13 (36.1) | 12 (33.3) | 11 (30.6) |
| 6. All individuals should be assessed on admission to a hospital for risk of PI development. | 30 (83.3) | 5 (13.9) | 1 (2.8) |
| 7. Patient skin should be clean and dry to prevent risk of PI development. | 32 (88.9) | 3 (8.3) | 1 (2.8) |
| 8.Adequate dietary intake of protein and calories should be maintained during illness. | 26 (72.2) | 2 (5.6) | 8 (22.2) |
| 9.Vitamin C & E are important to maintain skin integrity. | 29 (80.6) | 1 (2.8) | 6 (16.7) |
| 10.Persons confined to bed should be repositioned every three hours. | 24 (66.7) | 8 (22.2) | 4 (11.1) |
| 11.A person who cannot move him or herself should be repositioned every two hours while sitting in a chair. | 33 (91.7) | 3 (8.3) | 0 (0) |
| 12.Heel injury is prevented by Putting pillow under the patient's leg. | 32 (88.9) | 1 (2.8) | 3 (8.3) |
| 13.Friction may occur when moving a person up in bed. | 33 (88.9) | 1 (2.8) | 3 (8.3) |
| 14.For persons who have incontinence, skin cleaning should occur at the time of soiling and at routine intervals. | 34 (94.4) | 1 (2.8) | 1 (2.8) |
| 15.Educational programs may reduce the incidence of pressure injuries. | 33 (91.7) | 3 (8.3) | 0 (0) |
| 16.Stage II pressure injuries may be extremely painful due to exposure of nerve endings. | 23 (63.9) | 3 (8.3) | 10 (27.8) |
| 17.Shear is the force that occur when the skin sticks to the surface and the body slides. | 28 (77.8) | 2 (5.6) | 6 (16.7) |
| 18.All care given to prevent or treat PI must be documented. | 33 (91.7) | 0 (0) | 3 (8.3) |

Table 3 presents the attitudes of physiotherapists towards PI prevention, reflecting a wide range of responses among physiotherapists regarding the prevention of PI. They mostly agree that continuous assessment of patients will give an accurate account of their PI risk, with 52.8%. It reflects a positive attitude level toward preventing PI from this issue. Otherwise, 41% agree or disagree with the statement "My clinical judgment is better than any PI risk assessment tool available to me", which reflects poor attitudes towards PI prevention protocols.

Table 4 shows physiotherapists' perceived barriers towards PI. Shortage of resources was perceived most in the medical ward as 94.4% acknowledge it, lack of multidisciplinary team among the health care team, and (88.9) and uncooperative patients (88.9%) followed by presence of other priorities at work than PI (83.3%), and in adequate knowledge about PI (80.6%). However, the remaining barriers were acknowledged by one-third of the participants as poor access to literature, a heavy workload, inadequate staff, and a lack of universal guidelines.

As shown in Table 5, gender and marital status were not found to be significantly associated with scores for knowledge, attitude, or perceived barriers. Marital Status was not statistically significant in relation to knowledge, attitudes, or barriers. A significant finding emerged for age, which was the only factor related to attitudes. Older participants demonstrated progressively more positive attitudes (F = 6.8, P < 0.001**). Age was not, however, significantly related to knowledge or barrier scores.. In contrast, knowledge scores were significantly associated with several factors. Participants with a monthly income greater than 500 JD had significantly higher knowledge (t = 3.55, P = 0.01**), though income did not significantly impact attitude or barrier scores. While in terms of Work Experience, Knowledge scores varied significantly across different levels of work experience, with participants having ≤ 5 years scoring 12.0, those with 6–10 years scoring 16.3, those with 11–15 years scoring 14.1, and those with > 15 years scoring 14.8 (F = 3.47, P = 0.03*). Attitude scores did not show significant differences (F = 2.34, P = 0.10), nor did perceived barriers (F = 1.67, P = 0.19). Participants who received

**Table 3. Physiotherapist's attitude towards preventing PI (n = 36).**

| Variables | Physiotherapist's Attitude | | | | |
|---|---|---|---|---|---|
| | Strongly disagree n(%) | Disagree n (%) | neither agree nor disagree n(%) | Agree n (%) | Strongly agree n (%) |
| All patients are at potential risk of developing PI. | 1 (2.8) | 6 (16.7) | 11 (30.6) | 12 (33.3) | 6 (16.7) |
| PI prevention is time consuming for me to carry out. | 4 (11.1) | 7 (19.4) | 6 (16.7) | 16 (44.4) | 3 (8.3) |
| Pressure injury treatment is a greater priority than pressure injury prevention. | 3 (8.3) | 4 (11.1) | 16 (44.4) | 7 (19.4) | 6 (16.7) |
| Continuous assessment of patients will give an accurate account of their PI risk. | 1 (2.8) | 3 (8.3) | 5 (13.9) | 19 (52.8) | 8 (22.2) |
| Most PI can be avoided. | 0 (0) | 5 (13.9) | 14 (38.9) | 11 (30.6) | 6 (16.7) |
| In comparison with other areas of care, PI Prevention is a low priority for me. | 3 (8.3) | 6 (16.7) | 13 (33.3) | 11 (30.6) | 4 (11.1) |
| PI risk assessment should be regularly carried out on all patients during their stay in hospital. | 0 (0) | 6 (16.7) | 5 (13.9) | 16 (44.4) | 9 (25.0) |
| I do not need to concern myself with pressure injury prevention in my practice. | 6 (16.7) | 8 (22.2) | 13 (36.1) | 8 (22.2) | 1 (2.8) |
| I am less interested in PI prevention than other aspects of care. | 7 (19.4) | 5 (13.9) | 9 (25.0) | 13 (36.1) | 2 (5.6) |
| In my opinion, patients tend not to get as many PI nowadays. | 1 (2.8) | 4 (11.1) | 18 (50.0) | 6 (16.7) | 7 (19.4) |
| My clinical judgment is better than any PI risk assessment tool available to me. | 4 (11.1) | 3 (8.3) | 14 (38.9) | 11 (30.6) | 4 (11.1) |

**Table 4. Physiotherapists' perceived barriers towards preventing PI (n = 36).**

| Barriers | Physiotherapist's Perceived Barriers | |
|---|---|---|
| | Yes n (%) | No n (%) |
| Poor access to literature and reading facilities. | 28 (77.8) | 8 (22.2) |
| Heavy workload and inadequate staff. | 26 (72.2) | 10 (27.8) |
| Lack of universal guideline on prevention of PI. | 26 (72.2) | 10 (27.8) |
| Inadequate training coverage of pressure injury prevention. | 28 (77.8) | 8 (22.2) |
| Uncooperative patients. | 32 (88.9) | 4 (11.1) |
| Lack of job satisfaction in your profession. | 26 (72.2) | 10 (27.8) |
| Presence of other priorities than pressure injury. | 30 (83.3) | 6 (16.7) |
| Shortage of resources (equipment/resource). | 34 (94.4) | 2 (5.6) |
| Inadequate knowledge about PI among health care providers. | 29 (80.6) | 7 (19.4) |
| Lack of multidisciplinary among health care team. | 32 (88.9) | 4 (11.1) |
| I don't have any challenge. | 15 (41.7) | 21 (58.3) |

training (n = 15) had a higher mean knowledge score (15.6) compared to those who did not receive training (13.3), with a significant difference (t = 3.35, P = 0.02*). Those who followed guidelines (n = 22) had a significantly higher mean knowledge score (15.3) compared to those who did not (12.7, t = 3.89, P < 0.001**). The unproportioned provider ratio shows no significant difference in knowledge, attitude, and barriers scores.

Table 6 presents the results of a multivariate regression analysis examining factors associated with knowledge, attitude, and perceived barriers regarding PI among health providers. Significant predictors of knowledge included monthly income (β = 1.49, P = 0.03*) and the use of guidelines (β = −1.80, P = 0.02*). In contrast, age was a significant predictor of attitude

**Table 5. Comparing knowledge, attitude, and perceived barrier scores results by demographic information (n = 36).**

| Demographic information | Knowledge | | | Attitude | | | Perceived barriers | | |
|---|---|---|---|---|---|---|---|---|---|
| | Mean | Test | P | Mean | Test | P | Mean | Test | P |
| Gender | | | | | | | | | |
| ● Male (n = 22) | 13.9 | t = 1.07 | 0.29 | 35.5 | t = 1.33 | 0.19 | 8.9 | t = 1.41 | 0.1 |
| ● Female (n = 14) | 14.8 | | | 33.4 | | | 7.9 | | 7 |
| Marital Status | | | | | | | | | |
| ● Single (8) | 13.6 | t = 0.92 | 0.37 | 33.4 | t = 0.86 | 0.39 | 9.3 | t = 1.16 | 0.25 |
| ● Married (28) | 14.5 | | | 35.0 | | | 8.3 | | |
| Age | | | | | | | | | |
| ● < 30 (n = 7) | 13.1 | F = 1.43 | 0.26 | **30.6** | **F = 6.8** | **0.001**** | 9.7 | F = 1.59 | 0.22 |
| ● 30-40 (n = 22) | 14.4 | | | **34.7** | | | 8.1 | | |
| ● > 40 (n = 7) | 15.1 | | | **38.6** | | | 8.4 | | |
| Monthly Income | | | | | | | | | |
| ● ≤ 500 JD (n = 18) | **13.1** | **t = 3.55** | **0.01 **** | 34.5 | t = 0.18 | 0.86 | 7.9 | t = 1.6 | 0.11 |
| ● > 500 JD (n = 18) | **15.4** | | | 34.8 | | | 9.1 | | |
| Work Experience | | | | | | | | | |
| ● ≤ 5 yr (n = 5) | **12.0** | **F = 3.47** | **0.03*** | 30.4 | F = 2.34 | 0.10 | 9.6 | F = 1.67 | 0.19 |
| ● 6 − 10 yr (4) | **16.3** | | | 36.3 | | | 9.0 | | |
| ●11–15 yr (15) | **14.1** | | | 34.3 | | | 8.8 | | |
| ● > 15 yr (n = 12) | **14.8** | | | 36.3 | | | 7.5 | | |
| Training | | | | | | | | | |
| ● Yes (n = 15) | **15.6** | **t = 3.35** | **0.02 *** | 35.1 | t = 0.46 | 0.65 | 9.1 | t = 1.4 | 0.17 |
| ● No (n = 21) | **13.3** | | | 34.3 | | | 8.1 | | |
| Guidelines | | | | | | | | | |
| ● Yes (n = 22) | **15.3** | **t = 3.89** | **0.001**** | 33.8 | t = 1.41 | 0.17 | 8.1 | t = 1.51 | 0.14 |
| ● No (n = 14) | **12.7** | | | 36.0 | | | 9.1 | | |
| Unproportional Providers rati o | | | | | | | | | |
| ● Yes (n = 18) | 15.6 | t = 0.73 | 0.47 | 34.3 | t = 0.39 | 0.70 | 8.3 | t = 0.48 | 0.64 |
| ● No (n = 18) | 14.0 | | | 34.9 | | | 8.7 | | |

\* Significant at P < 0.05, \*\* Significant at P < 0.01.

(β = 5.37, P < 0.001\*\*). The model statistics indicate that the knowledge model explained 59% of the variance, while the attitude model explained 45%, and the perceived barriers model explained 36%, with the knowledge and attitude models showing a significant P value (0.001\*\* and 0.02\*, respectively). Overall, guideline use and prior PI training were associated with higher knowledge.

## Discussion

The current study aims to assess the knowledge, attitudes, and barriers among physiotherapists regarding the prevention of PI in medical wards. The overall findings revealed that physiotherapists possessed a good knowledge base and positive attitudes toward PI prevention. This aligns with a study conducted in Saudi Arabia at a rehabilitation center, where physiotherapists demonstrated good knowledge and attitudes compared to other healthcare professions [27] It is worth mentioning that rehabilitation centers in Jordan treat patients with chronic medical conditions and provide palliative care, which is similar to the care offered in medical wards. Additionally, this result supports the study by Hamdan and colleagues, who found that healthcare providers' attitudes toward PI prevention in the cancer unit in Saudi Arabia were highly

**Table 6. Multivariate regression analysis of factors associated with knowledge, attitude, and perceived barriers about PI among health providers.**

| Independent Variables | Knowledge | | | | | Attitude | | | | | Perceived Barriers | |
|---|---|---|---|---|---|---|---|---|---|---|---|---|
| | (β) | STD (β) | 95% CI for (β) | | P-Value | (β) | STD (β) | 95% CI for (β) | | P-Value | (β) | P-Value |
| | | | Lower | Upper | | | | | Lower | | | Upper |
| Gender | 1.12 | 0.24 | −.23 | 2.46 | 0.09 | −0.11 | −0.01 | −3.31 | 3.10 | 0.95 | −0.53 | 0.48 |
| Age | 0.51 | 0.14 | −1.05 | 2.08 | 0.51 | **5.37** | 0.73 | 1.64 | 9.11 | **0.001 **** | 0.09 | 0.92 |
| Marital Status | 0.17 | −0.03 | −1.92 | 1.57 | 0.84 | −1.49 | −0.13 | −5.64 | 2.67 | 0.47 | −0.29 | 0.76 |
| Monthly Income | **1.49** | **0.33** | **.18** | **2.79** | **0.03 *** | 0.09 | 0.01 | −3.01 | 3.21 | 0.95 | 1.29 | 0.08 |
| Work Experience | −0.03 | −0.01 | −1.09 | 1.04 | 0.96 | −0.99 | −0.22 | −3.54 | 1.56 | 0.43 | −0.79 | 0.19 |
| Training | −1.19 | −0.26 | −2.53 | 0.15 | 0.08 | −0.65 | −0.07 | −3.85 | 2.55 | 0.68 | −0.87 | 0.25 |
| Using Guidelines | **−1.80** | **−0.39** | **−3.29** | **−0.31** | **0.02 *** | **4.27** | 0.45 | .71 | 7.84 | **0.02 *** | 1.54 | 0.07 |
| Unproportional providers | 0.69 | .154 | −0.64 | 2.02 | 0.30 | −1.31 | −0.14 | −4.474 | 1.85 | 0.40 | −0.41 | 0.58 |
| Model Statistics | | | | | | | | | | | | |
| ● R2 | 0.59 | | | | | 0.45 | | | | | 0.36 | |
| ● F Statistics | 4.78 | | | | | 2.71 | | | | | 1.93 | |
| ● P value | 0.001 ** | | | | | 0.02 * | | | | | 0.09 | |

\* Significant at P<0.05, ** Significant at P<0.01; STD (β): Standardized Coefficients.

positive [4], similar to Asif et al.'s findings on physiotherapists' satisfactory knowledge and positive attitudes [13]. However, the study also observed that at one point in their attitudes, physiotherapists showed less interest in PI prevention during care. This may be related to barriers that negatively affect their attitude, such as focusing on other clinical priorities, high workloads, and limited time for direct PI patient care related to prevention. Moreover, cooperation from PI patients and support from family members may be insufficient, leading to decreased perceived effectiveness and frustration with PI prevention efforts. These outcomes are consistent with perceived barriers identified by [4], who reported negative attitudes in similar settings due to environmental and systemic challenges.

Our study found that physiotherapists perceived several barriers to PI prevention, including a lack of resources and supplies in the medical wards. This problem was highlighted in several studies, as found in a recent literature review study [28] in general hospital settings. However, in the medical ward, this might be a significant challenge, as limited resources are available in medical wards in Jordan. Most resources for preventing PI are likely to be focused on critical care units rather than medical wards, which requires considerable policy attention and fair resource distribution. On the other hand, inadequate knowledge of PI prevention was the other recognized barrier, as although the majority of them had received old training, no updated knowledge or no training was given to them, particularly working on medical wards, as was found in the literature review study [28]. In addition to uncooperative patients, who required a lot of attention and work among the other healthcare providers, team members faced the barriers listed as well. Patients in the medical ward mainly complain of stroke or other comorbidities, which can increase the incidence rate of PI [29]. Therefore, family members need to be educated on the PI prevention, and special care might be required; this might need more family health education and empowerment for early intervention and proper PI management [17,30]. The Physiotherapists also acknowledge that the priorities of their work might be to focus more on other than PI prevention since working on the medical wards with high workload and shortage of staff and overcrowding at the governmental hospitals might affect their priorities and attention as well, which was found in the study of Azhar et al. and Mkoka & Andwilile studies [31,32].

Our study highlighted significant factors related to the physiotherapist's knowledge, attitudes, and barriers scores. In terms of knowledge, our analysis reveals that monthly income, work experience, training, and the use of guidelines were significantly related to good knowledge of PI. Monthly income might be a critical factor in the knowledge score, as those who receive higher incomes may have higher qualifications or more work experience, based on the government income scale, which is both related to better knowledge and experience in working with such cases. This was highlighted in the study of Ozcon and his colleagues in Turkey [18]. They also emphasized the importance of training in their study, where they found that a low number of participants received training for PI prevention, which could impact their knowledge of PI prevention. In our research, training and use guidelines were significantly associated with good knowledge, as using guidelines and training are highly important factors in improving physiotherapists' knowledge regarding PI prevention [28].

In terms of attitude scores, it was significantly related to the age variable in our study. Age might play an essential role in attitudes in general, as positive attitudes regarding PI prevention might be related to the importance of this health outcome on patients, and being more empathetic and willing to help, and decreasing the problem of this health outcome problem [33].

The barriers perceived by physiotherapists were not significantly related to any demographic characteristics, as most of the barriers were associated with systemic problems or organizational issues, such as resource limitations, excessive workloads, or staff shortages, which have also been highlighted in other studies. However, these challenges may be exacerbated by the specific conditions of the medical wards [28].

The multidisciplinary team approach was one of the recognized barriers that the physiotherapists acknowledged in our study. Where the medical ward has special needs and further attention. Collaboration among physicians, wound care nurses, registered nurses, physiotherapists, specialists, and other healthcare providers is significant for improving knowledge and attitudes towards PI prevention. Cohesive teamwork can raise the practical encountered barriers to best PI prevention practices, support each other, define roles and responsibilities, and share experiences in solving challenges [34]. Physiotherapists role goes beyond rehabilitation and physical activities, it can play a significant role in PI prevention in medical wards same as ICU and critical care unit when worked collaboratively with the other healthcare team [35].

## Conclusions

Physiotherapists play a vital role and take significant responsibility in preventing PI. Our study indicates that while there is satisfactory knowledge and a positive attitude towards PI prevention, many systemic and organizational barriers have been identified that require policy decisions and organizational changes, such as increasing the number of staff assigned to medical wards and providing all necessary resources for these wards. Training and guidelines were linked to the physiotherapists' good knowledge; therefore, specialized training that emphasizes updated protocols and guidelines is highly recommended to enhance their knowledge and ultimately improve patient health status, quality of care, and patient safety. Moreover including PI prevention in the physiotherapy curriculum would prepare the new graduates for Pi prevention as well.

Our study strongly advocates for future intervention research on a multidisciplinary team approach in both managing and preventing PI, which could significantly reduce the incidence of PI among patients in medical wards, improve overall health outcomes, and decrease healthcare costs.

## Strengths and limitations

The study assessed the level of knowledge, attitudes, and barriers among physiotherapists working in the largest government hospitals in Jordan regarding the PI in medical wards. Given the high demand for care and PI prevention in these crowded settings, this area remains under-researched. The small sample size is one of the acknowledged limitations of this study, despite using a convenience sample that included all physiotherapists currently employed in these hospitals and specific wards. While a larger sample could enhance the strength and generalizability of the findings, this limitation is

acknowledged. Despite this, the study provides valuable insights, particularly given the unique challenges faced in Jordan's overcrowded medical wards. The results can help inform policymakers in initiating training programs, developing targeted guidelines, assigning specific roles for healthcare providers, and enhancing a multidisciplinary team approach. Although the sample size may somewhat limit the generalizability of the results, the study makes a significant contribution to our understanding of PI prevention in medical wards. It highlights the need for further research with a larger sample and covering several health care settings in this crucial area.

## Acknowledgments

The researcher would like to acknowledge the participants of the study for their time and dedication to participation. Special thanks are given to the administrators in the medical wards for their valuable cooperation. The authors are also grateful to the Deanship of Scientific Research, King Khalid University, for its support of this work.

## Author contributions

**Conceptualization:** Enas A Assaf, Rahaf Alkhresheh, Taher Assaf, Suhair Al-Ghabeesh, Ramasubbamma Ramaiah.

**Data curation:** Enas A Assaf, Rahaf Alkhresheh.

**Formal analysis:** Taher Assaf.

**Funding acquisition:** Ramasubbamma Ramaiah.

**Investigation:** Enas A Assaf, Rahaf Alkhresheh.

**Methodology:** Enas A Assaf, Rahaf Alkhresheh, Taher Assaf.

**Project administration:** Enas A Assaf.

**Supervision:** Enas A Assaf, Rahaf Alkhresheh.

**Writing – original draft:** Enas A Assaf, Rahaf Alkhresheh, Taher Assaf, Suhair Al-Ghabeesh, Ramasubbamma Ramaiah.

**Writing – review & editing:** Enas A Assaf, Taher Assaf, Suhair Al-Ghabeesh, Ramasubbamma Ramaiah.

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
