## [Decision Letter · Decision Letter 0]

23 Oct 2025

Dear Dr. Assaf,

Thank you for submitting your manuscript to PLOS ONE. After careful consideration, we feel that it has merit but does not fully meet PLOS ONE’s publication criteria as it currently stands. Therefore, we invite you to submit a revised version of the manuscript that addresses the points raised during the review process.

We look forward to receiving your revised manuscript.

Kind regards,

Maheshkumar Baladaniya

Academic Editor

PLOS ONE

Journal Requirements:

This research was funded by the Deanship of Scientific Research at King Khalid University; Grant number "RGP 2/263/46".

3. Please amend the manuscript submission data (via Edit Submission) to include author Al-Ghabeesh S

4. Please amend your authorship list in your manuscript file to include author suhair Al-Ghabeesh

Additional Editor Comments:

Need to make changes as advised by reviewers.

Reviewers' comments:

Reviewer's Responses to Questions

**Comments to the Author**

1. Is the manuscript technically sound, and do the data support the conclusions?

Reviewer #1: Yes

Reviewer #2: Yes

2. Has the statistical analysis been performed appropriately and rigorously?

Reviewer #1: Yes

Reviewer #2: Yes

3. Have the authors made all data underlying the findings in their manuscript fully available?

Reviewer #1: Yes

Reviewer #2: Yes

4. Is the manuscript presented in an intelligible fashion and written in standard English?

Reviewer #1: Yes

Reviewer #2: Yes

Reviewer #1: Overall Assessment

The topic is timely and relevant given the growing emphasis on multidisciplinary approaches in healthcare and the scarcity of data from Middle Eastern contexts. However, it would benefit from further refinement in the discussion, statistical interpretation, and contextualization of findings within broader global and regional literature.

Introduction

The introduction could better emphasize the research gap—why physiotherapists in medical wards (as opposed to ICUs or rehabilitation units) have been overlooked.

It could benefit from a clear statement of study objectives or hypotheses at the end.

“Therefore, this study aimed to assess physiotherapists’ knowledge, attitudes, and perceived barriers toward pressure injury prevention in Jordanian medical wards and identify associated demographic factors.”

Methods

Convenience sampling of 36 physiotherapists limits external validity; this limitation should be discussed earlier in the manuscript.

The statistical analysis section needs a brief mention of how missing data (if any) were handled.

Clarify whether the regression model controlled for confounders or if it was unadjusted.

Results

The results section sometimes repeats data already visible in tables. Summaries could be more interpretive rather than descriptive.

Ensure consistency between reported p-values (e.g., 0.00 should be written as p < 0.001).

Discussion

The discussion lacks a multidisciplinary systems perspective—for example, collaboration between physiotherapists, nurses, and wound care teams could be more deeply analyzed.

Broader international comparison (outside the Middle East) would strengthen the global relevance.

Recommend acknowledging limitations related to social desirability bias and self-reported measures.

The discussion section comprehensively interprets findings but lacks a broader multidisciplinary perspective linking physiotherapists’ preventive and rehabilitative roles to overall patient well-being. Incorporating literature highlighting the evolving preventive and psychosocial dimensions of physiotherapy would strengthen the scientific depth and contextual relevance of the discussion, emphasizing physiotherapists’ integral role in holistic pressure injury prevention. Add below phrase.

“The evolving role of physiotherapists extends beyond rehabilitation, encompassing preventive and psychosocial health dimensions [Nasif et al., 2025 https://doi.org/10.31579/2578-8868/359 ]. Such multidisciplinary engagement supports patient-centered approaches essential in pressure injury prevention.”

Conclusion and Implications

The conclusion could benefit from more specific, evidence-backed recommendations (e.g., integrating PI prevention modules into physiotherapy curricula). Add a sentence emphasizing the study’s contribution to regional literature and how it can inform future interventional research.

Reviewer #2: Methods:

Sample size: The rationale provided through GPower appears to be at odds with the final figure 36 participants. Determine if this represented the entire eligible population or if difficulties in recruitment restricted participation.

Instrument: The process of adapting the questionnaire is clearly outlined, yet additional information is required about translation, cultural adjustments, and pretesting specific to the Jordanian setting.

Data Analysis: While statistical tests like t-test, ANOVA, regression are appropriate, the assumptions normality, multicollinearity are not addressed. Think about adding confidence intervals and effect sizes for important findings.

Result:

Explain the application of percentages compared to raw numbers to guarantee uniformity among tables.

The regression analysis is clearly displayed but would improve by including standardized beta coefficients and confidence intervals.

Certain p-values are presented as 0.00, they ought to be stated as < 0.001.

Discussion:

Elaborate on the implications for policy and clinical training what targeted actions can organizations implement to enhance multidisciplinary pressure injury prevention?

Add a brief section discussing how this research contributes to the current literature and suggest areas for future studies, such as intervention trials and larger national samples.

Explicitly recognize the limitations of the small sample size, the singular setting, and the self-reporting bias.

Result:

Add a more emphatic statement linking enhanced knowledge and guideline application by physiotherapists to measurable patient-care results.

Refrain from including new details or references to literature in this section

**Do you want your identity to be public for this peer review?** For information about this choice, including consent withdrawal, please see our Privacy Policy

Reviewer #1: No

Reviewer #2: No

---

## [Author Response · Author response to Decision Letter 1]

21 Nov 2025

Please ensure that your manuscript meets PLOS ONE's style requirements, including those for file naming. The PLOS 1. ONE style templates can be found at

done accordingly

This research was funded by the Deanship of Scientific Research at King Khalid University; Grant number "RGP 2/263/46".

included in the cover letter

3. . Please amend the manuscript submission data (via Edit Submission) to include author Al-Ghabeesh S

4. Please amend your authorship list in your manuscript file to include author Suhair Al-Ghabeesh

done

Done.

yes, done

All references were checked one by one for retraction, and no retractions were reported either on the journal page or Google Scholar, however, I found one reference irrelevantly inserted and removed. Thank you for this important note.

reviewer one

Overall Assessment

The topic is timely and relevant given the growing emphasis on multidisciplinary approaches in healthcare and the scarcity of data from Middle Eastern contexts. However, it would benefit from further refinement in the discussion, statistical interpretation, and contextualization of findings within broader global and regional literature.

Introduction

The introduction could better emphasize the research gap—why physiotherapists in medical wards (as opposed to ICUs or rehabilitation units) have been overlooked.

It could benefit from a clear statement of study objectives or hypotheses at the end.

Thank you for your comments:

Added more global and international literature to both the introduction and the discussion

Introduction, the study aim was changed and we took your clearer statement. Thank you

Methods

Convenience sampling of 36 physiotherapists limits external validity; this limitation should be discussed earlier in the manuscript.

The statistical analysis section needs a brief mention of how missing data (if any) were handled.

Clarify whether the regression model controlled for confounders or if it was unadjusted.

a. A convenient sample limitation had been added in the method section

b. In the study, we collected data from all 36 physiotherapists targeted in the three hospitals. During data cleaning, it was confirmed that all 36 returned questionnaires were fully completed. We added a sentence to the "Statistical analysis" section to clarify this.

c. The multivariate regression models were adjusted. As described in the "Statistical analysis" section and presented in Table 6, all independent variables (demographic factors: gender, age, marital status, monthly income, work experience, training, using guidelines, and unproportional providers) were entered simultaneously into the regression models for each of the three outcome variables. Statistical analysis section were modified accordingly.

Results

The results section sometimes repeats data already visible in tables. Summaries could be more interpretive rather than descriptive.

Ensure consistency between reported p-values (e.g., 0.00 should be written as p < 0.001).

Done. We have revised the results text for Table 5 to be more interpretive and less descriptive. We also corrected all 'P = 0.00' values to 'P < 0.001' in the text and tables for statistical consistency

Discussion

The discussion lacks a multidisciplinary systems perspective—for example, collaboration between physiotherapists, nurses, and wound care teams could be more deeply analyzed.

Broader international comparison (outside the Middle East) would strengthen the global relevance.

Recommend acknowledging limitations related to social desirability bias and self-reported measures.

The discussion section comprehensively interprets findings but lacks a broader multidisciplinary perspective linking physiotherapists’ preventive and rehabilitative roles to overall patient well-being. Incorporating literature highlighting the evolving preventive and psychosocial dimensions of physiotherapy would strengthen the scientific depth and contextual relevance of the discussion, emphasizing physiotherapists’ integral role in holistic pressure injury prevention. Add below phrase.

“The evolving role of physiotherapists extends beyond rehabilitation, encompassing preventive and psychosocial health dimensions [Nasif et al., 2025 https://doi.org/10.31579/2578-8868/359 ]. Such multidisciplinary engagement supports patient-centered approaches essential in pressure injury prevention.”

Discussed further and highlighted the importance of the physiotherapist and the ref. were added for it's significant

Conclusion and Implications

The conclusion could benefit from more specific, evidence-backed recommendations (e.g., integrating PI prevention modules into physiotherapy curricula). Add a sentence emphasizing the study’s contribution to regional literature and how it can inform future interventional research.

this was added

Reviewer two

Methods:

Sample size: The rationale provided through GPower appears to be at odds with the final figure 36 participants. Determine if this represented the entire eligible population or if difficulties in recruitment restricted participation.

Instrument:

The process of adapting the questionnaire is clearly outlined, yet additional information is required about translation, cultural adjustments, and pretesting specific to the Jordanian setting.

Data Analysis: While statistical tests like t-test, ANOVA, regression are appropriate, the assumptions normality, multicollinearity are not addressed. Think about adding confidence intervals and effect sizes for important findings.

Sample size: The sample size were the entire physiotherapist working in the three studied hospitals, as they were all interested in participating in the study; moreover, those physiotherapists were assigned to cover the entire hospital department as well. Therefore, During the study period, the number of physiotherapists meeting our eligibility criteria (working in medical wards at the three participating governmental hospitals) was finite. We therefore adopted a census approach and invited all eligible physiotherapists. The final sample of 36 represents the entire eligible population available at that time; consequently, the a-priori GPower calculation served only as initial planning guidance and was superseded by the census design. We have clarified this in the Methods.

Instruments

This was added as the questionnaire is written and administered in English. And a pilot study was conducted as well

We revised Table 6 to report, for the knowledge and attitude outcomes, the adjusted unstandardized coefficients (B), 95% confidence intervals, and standardized coefficients (β), alongside model fit statistics; p-values are consistently formatted. The perceived barriers model did not reach overall significance (F = 1.93, p = .09), so to avoid over interpretation we kept its presentation parsimonious in the main table; we are happy to add β and 95% CIs for barriers in a supplement if preferred. (See revised Table 6.)

Result:

Explain the application of percentages compared to raw numbers to guarantee uniformity among tables.

The regression analysis is clearly displayed but would improve by including standardized beta coefficients and confidence intervals.

Certain p-values are presented as 0.00, they ought to be stated as < 0.001.

Across all descriptive tables, categorical variables are reported uniformly as count and percentage [n (%)]. Percentages are consistently calculated using the total sample size (N = 36) as the denominator because there were no missing data for the variables presented.

Discussion:

Elaborate on the implications for policy and clinical training what targeted actions can organizations implement to enhance multidisciplinary pressure injury prevention?

Add a brief section discussing how this research contributes to the current literature and suggest areas for future studies, such as intervention trials and larger national samples.

Explicitly recognize the limitations of the small sample size, the singular setting, and the self-reporting bias.

Done, the recommendations were further elaborated focusing multidisciplinary approach in future studies for improving PI prevention

Sample size limitation was acknowledged and discussed in the limitations section

Result:

Add a more emphatic statement linking enhanced knowledge and guideline application by physiotherapists to measurable patient-care results.

Refrain from including new details or references to literature in this section

In the Results, we added a concluding sentence to the regression summary: “Overall, guideline use and prior PI training were associated with higher knowledge.” This emphasizes the observed relationship using only our data.

---

## [Decision Letter · Decision Letter 1]

23 Feb 2026

"Healing Hands: Understanding Physiotherapists' Knowledge, Attitudes and Barriers on Pressure Injury Prevention in the Medical Wards in Jordan"

PONE-D-25-37371R1

Dear Dr. Assaf,

We’re pleased to inform you that your manuscript has been judged scientifically suitable for publication and will be formally accepted for publication once it meets all outstanding technical requirements.

Kind regards,

Maheshkumar Baladaniya

Academic Editor

PLOS One

Additional Editor Comments (optional):

Manuscript it acceptable for publication.

Reviewers' comments:

Reviewer's Responses to Questions

**Comments to the Author**

Reviewer #1: All comments have been addressed

Reviewer #2: All comments have been addressed

2. Is the manuscript technically sound, and do the data support the conclusions?

Reviewer #1: Yes

Reviewer #2: Yes

3. Has the statistical analysis been performed appropriately and rigorously?

Reviewer #1: Yes

Reviewer #2: Yes

4. Have the authors made all data underlying the findings in their manuscript fully available?

Reviewer #1: Yes

Reviewer #2: Yes

5. Is the manuscript presented in an intelligible fashion and written in standard English?

Reviewer #1: Yes

Reviewer #2: Yes

Reviewer #1: After re-evaluation of the revised manuscript and the detailed point-by-point responses, I find that the authors have effectively incorporated all suggested changes. The revisions have strengthened the manuscript, and it is now ready for acceptance.

Reviewer #2: The authors have adequately responded to reviewer comments, and the revised version is scientifically sound, clearly written, and suitable for publication in PLOS ONE.

**Do you want your identity to be public for this peer review?** For information about this choice, including consent withdrawal, please see our Privacy Policy

Reviewer #1: No

Reviewer #2: No

---

## [Editor Report · Acceptance letter]

PONE-D-25-37371R1

PLOS One

Dear Dr. Assaf,

I'm pleased to inform you that your manuscript has been deemed suitable for publication in PLOS One. Congratulations! Your manuscript is now being handed over to our production team.

Kind regards,

on behalf of

Dr. Maheshkumar Baladaniya

Academic Editor

PLOS One